# CoCMT: Towards Communication-Efficient Cross-Modal Transformer for Collaborative Perception

## Abstract

Cooperative perception systems in autonomous driving enhance each agent's perceptual capabilities by sharing visual information with others and demonstrated effectiveness in handling prominent challenges like occlusions and long-range detection. However, most existing cooperative systems transmit feature maps, such as bird's-eye view (BEV) representations, which include substantial background data and are costly to process due to their high dimensionality. This paradigm introduces a trade-off between improved perception and increased communication overhead. To address this challenge, we present CoCMT, an object-query-based collaboration framework that enables efficient communication while unifying homogeneous and heterogeneous cooperative perception tasks. Within CoCMT, we introduce the Efficient Query Transformer (EQFormer) to effectively fuse multi-agent object queries and implement a synergistic deep supervision approach to accelerate convergence during training. Extensive experiments on the OPV2V and V2V4Real datasets demonstrate that CoCMT surpasses current state-of-the-art methods in performance while offering significant communication efficiency. Notably, on the real-world V2V4Real dataset, our proposed CoCMT model (Top-50 object queries) requires merely 0.416 Mb bandwidth during inference. This reduces bandwidth consumption by 323 times compared to SOTA methods while improving AP@70 by 1.1. The code and models will be open-sourced.

## 1 Introduction

Accurate and efficient perception is essential for autonomous driving (AV) to ensure reliable navigation and safe decision-making. However, single-vehicle autonomous systems face significant challenges in real-world scenarios, such as occlusions and limited sensing range. Cooperative perception systems address these issues by enabling agents to enhance their perceptual capabilities through the sharing of sensing and visual information with other agents. Most research in cooperative perception systems (Xu et al., 2022c;a; Wang et al., 2020; Xu et al., 2022b; Wei et al., 2024) has primarily focused on homogeneous multi-agent perception, where all agents utilize the same type of sensors—such as LiDAR, cameras, or radars. Recent studies (Xiang et al., 2023; Lu et al., 2024) have advanced into heterogeneous multi-agent perception, facilitating collaboration between agents equipped with diverse sensor types. This approach better reflects real-world conditions, significantly enhancing the adaptability of cooperative perception systems and expanding their potential applications and impact. However, a trade-off exists between communication efficiency and perception performance in cooperative perception systems (Hu et al., 2022): while intermediate fusion methods improve performance, they generally demand significant communication bandwidth, as compared to simpler late fusion approaches whereas only the detection results are shared across agents.

Most existing cooperative perception fusion methods Xu et al. (2022c;b); Wei et al. (2024) use feature maps—such as Bird's-Eye-View (BEV) features—as the medium for information transmission among agents. Feature maps often employ high-dimensional representations to extend perception range and enhance performance; however, this also increases communication bandwidth requirements. Moreover, feature maps represent the entire scene surrounding the vehicle, where dynamic, relatively sparse foreground objects are mixed with a large amount of static background information. Transmitting large amounts of background data offers minimal benefit to perception performance

while occupying significant bandwidth. To this end, existing methods Lu et al. (2024); Xiang et al. (2023) have to incorporate complex foreground extraction mechanisms to reduce the unnecessary information being shared, which inevitably increases model complexity. To further reduce the feature redundancy, Hu et al. (2022; 2024) has focused on selecting key parts of the feature map or adopting alternative representations to balance performance and communication efficiency.

This paper proposes a novel object-centric framework tailored for communication-efficient collaborative perception. The sparsity nature of query-based object representations Carion et al. (2020); Li et al. (2022a) has offered several advantages over prior feature map-based strategies: 1) Small data size: The data size of object queries is significantly smaller than that of (BEV) feature maps, which can largely reduce the communication bandwidth required for transmission. 2) Object-centric focus: Unlike feature maps, which contain extensive background information, object queries are explicitly *object-centric*, encapsulating only the relevant contextual features and naturally excluding irrelevant background data. This eliminates the need for intermediate fusion algorithms to design complex foreground information extraction mechanisms (Hu et al., 2022; 2024). 3) Modality independence: Object queries are less dependent on specific data modalities, making them more versatile and effective for heterogeneous multi-agent perception tasks. *These advantages make object queries a more efficient and scalable choice for cooperative perception systems, especially in bandwidth-constrained and multi-modal environments.*

However, integrating object queries from multiple agents introduces two challenges. Firstly, object query-based models generate numerous initial queries, many of which are unrelated to actual objects. The challenge lies in efficiently filtering out noisy queries to merely focus on high-quality object queries for fusion. Moreover, object queries are unordered, meaning adjacent queries in the sequence may represent distant objects, especially when integrated from multiple agents. This unordered nature can cause feature confusion, complicating the interaction between relevant objects. To address these challenges, we introduce the **CoCMT** framework—**Co**mmunication-Efficient **C**ross-**M**odal **T**ransformer for Collaborative Perception. This framework utilizes object query as the medium for information transmission, effectively handling both homogeneous and heterogeneous multi-agent perception tasks using a unified and concise architecture. The framework is divided into two stages: the single-agent independent prediction stage and the cooperative fusion prediction stage. Additionally, we propose a synergistic deep supervision mechanism that applies deep supervision across both stages simultaneously, accelerating convergence and enhancing positive interactions between stages to improve overall performances. Extensive experiments on both simulated and real datasets demonstrate that our model achieves superior performance compared to State-of-the-art methods while requiring order-of-magnitude smaller communication bandwidth. Our contributions are:

- We propose CoCMT, a novel object query-based collaborative perception framework that uses object queries as intermediaries for information transmission, significantly reducing bandwidth consumption while enhancing the efficiency of collaborative perception.

- We design the Efficient Query Transformer (EQFormer), which incorporates three masking mechanisms to limit interactions between object queries to spatially valid, proximate, and strongly target-associated areas, ensuring precise and efficient attention learning for fusion.

- We introduce a Synergistic Deep Supervision mechanism that applies deep supervision at both the individual prediction and collaborative fusion stages. This mechanism accelerates model convergence during training and improves overall performance.

- Our extensive experiments on the OPV2V and V2V4real datasets validate the bandwidth efficiency of our proposed framework. The results demonstrate that the framework significantly reduces bandwidth consumption while achieving superior performance. We also conducted comprehensive ablation studies to demonstrate the efficacy of each component in our model design.

## 2 RELATED WORKS

### 2.1 COOPERATIVE PERCEPTION SYSTEMS

Cooperative perception systems enable connected and automated vehicles to communicate with others, thus enjoying shared perception capabilities to handle occlusions and long-distance perception issues Wang et al. (2020). Among the three types of cooperative perception—early fusion, intermediate, and late fusion—recent research has primarily focused on intermediate fusion methods, which

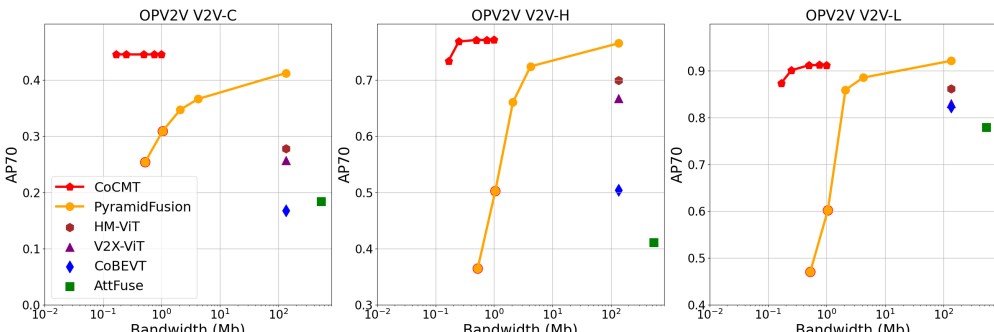

Figure 1: **AP vs Bandwidth.** The figure shows the variation in AP70 performance of the model under different bandwidth conditions, evaluating three settings in the OPV2V dataset: V2V-C, V2V-L, and V2V-H. Our CoCMT model, with a significant bandwidth advantage, demonstrates performance comparable to or even better than state-of-the-art (SOTA) methods. Moreover, as the bandwidth gradually decreases, CoCMT exhibits only minor performance degradation, fully showcasing its excellent adaptability to bandwidth fluctuations.

aim to improve cooperative perception performance by fusing intermediate neural features (Xu et al., 2022c; Wang et al., 2020; Xu et al., 2022a;b; Li et al., 2024c). For instance, V2VNet (Wang et al., 2020) uses a graph neural network to fuse feature maps from different agents. AttFuse (Xu et al., 2022c) combines self-attention with a local graph to learn interactions between feature maps. CoBEVT (Xu et al., 2022a) employs a fused axial attention module (FAX) to model interactions across different perspectives and agents. V2X-ViT (Xu et al., 2022b) utilizes a vision transformer architecture with two specially designed attention mechanisms to fuse heterogeneous feature maps in V2X scenarios. HEAL Lu et al. (2024) proposes a multi-scale foreground-aware Pyramid Fusion network to conduct heterogeneous collaborative perception.

## 2.2 CHALLENGES IN COOPERATIVE PERCEPTION

Despite their advantages, cooperative perception systems introduce several challenges, such as heterogeneous feature fusion, domain gaps, communication delays, and limited communication bandwidth, to name a few. Many studies have focused on enhancing the robustness of multi-agent cooperative perception systems to maintain perception performance (Xu et al., 2022b; Xiang et al., 2023; Hu et al., 2024; Wei et al., 2024; Xu et al., 2023a; Li et al., 2023; 2024a;b). For instance, CoBEVFlow (Wei et al., 2024) enhances robustness to asynchronous communication by compensating for motion through BEV Flow. To handle broader heterogeneous multi-agent perception in real-world scenarios, HMViT (Xiang et al., 2023) integrates heterogeneous sensor features from connected vehicles using a heterogeneous 3D graph transformer. HEAL (Lu et al., 2024) utilizes a PyramidFusion architecture to fuse heterogeneous features in a multi-scale and foreground-aware manner, and reduces the training cost for adding new heterogeneous agents through backward alignment. S2R-ViT (Li et al., 2024d) introduces sim-to-real transfer learning to minimize the sim2real domain gap in collaborative perception systems.

To reduce communication bandwidth, Where2comm (Hu et al., 2022) adopts a spatial confidence-aware strategy to transmit only the most critical feature information. CodeFilling (Hu et al., 2024) approximates feature maps using codebook-based representations and selects key information through information filling, achieving an optimal balance between communication and performance. QUEST (Fan et al., 2024) explores the use of object query as the information carrier in V2X scenarios, reducing communication bandwidth. However, these studies are limited to camera only homogeneous perception, and its performance degrades considerably when reducing the threshold of transmitted queries. In this paper, we extend the object query-based approach to simultaneously handle homogeneous and heterogeneous multi-agent perception tasks involving both LiDAR and cameras, aiming to achieve better communication-performance trade-offs.

## 2.3 3D OBJECT DETECTION

3D object detection plays a critical role in autonomous driving perception systems, and has undergone rapid development. Early multi-view camera-based 3D object detection methods (Philion &

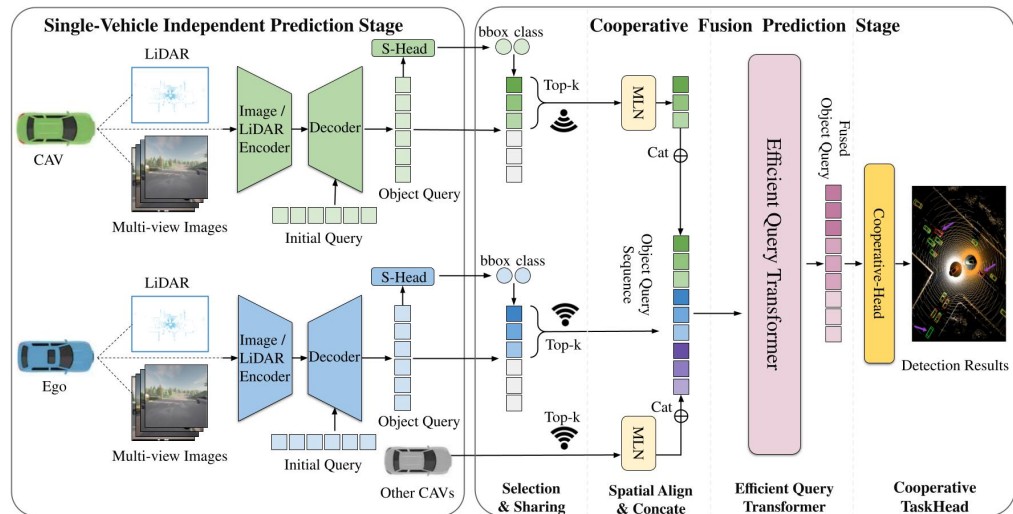

Figure 2: **Overview of the CoCMT framework.** This system consists of two stages: the single-agent independent prediction phase and the cooperative fusion prediction phase. The single-agent independent prediction stage can utilize any query-based 3D object detection model and it retaining the S-head (single-agent task head). The cooperative fusion prediction stage is composed of four key components: Information Selection and Sharing, Spatial Alignment and Concatenation, the Efficient Query Transformer, and Cooperative Taskheads. The MLN (Motion-aware Layer Normalization) is employed to perform spatial alignment for the object query.

Fidler, 2020; Li et al., 2022b; Liu et al., 2023) often relied on explicit view transformation or implicitly learned dense BEV features via Transformers to model the surrounding environment. To reduce dependence on complex view transformation processes, some work (Liu et al., 2022; Lin et al., 2022; 2023a;b; Yan et al., 2023; Wang et al., 2023) has explored sparse query techniques for efficiently sampling features. Especially, PETR (Liu et al., 2022) initializes object queries using 3D reference points, where these queries interact with 2D image features with added position embeddings within the Transformer decoder, directly learning spatial mappings from 2D to 3D. Sparse4D (Lin et al., 2022) leverages 4D key points to initialize object queries for sparse 4D key feature sampling. CMT (Yan et al., 2023) introduces the multi-modal 3D object detection framework by applying coordinate encoding for both image and point cloud features.

## 3 CoCMT Cooperative Perception Framework

We present CoCMT, illustrated in Figure 2, divided into two stages: the 1) single-agent prediction stage and the 2) cooperative fusion prediction stage. We adopt the standard query-based learning objective to train the single-agent perception. In the cooperative fusion prediction stage, we propose the Efficient Query Transformer (EQFormer) to restrict the interaction between object queries, achieved by applying several layers of attention masks. To accelerate the convergence of the framework and enhance the mutual reinforcement between the two stages, we propose a synergistic deep supervision mechanism that provides deep supervision for both stages simultaneously.

### 3.1 Single-Agent Independent Prediction Stage

In the first stage, we employ a query-based 3D object detection model to extract object queries, denoted as $Q_i \in \mathbb{R}^{N \times D}$, where $Q_i$ represents the set of object queries extracted from agent $i$. Each agent generates $N$ queries with $D$-dimensional features. We select $Q_i$ as the core intermediate features in the cooperative fusion prediction stage. Notably, unlike most cooperative perception models that rely solely on the backbone for feature extraction, our approach retains the task heads of the model at this stage. This retention allows us to incorporate additional predictive information—specifically, the 3D object centers $C_i \in \mathbb{R}^{N \times 3}$ and object class scores $S_i \in \mathbb{R}^{N \times C}$ —into the

subsequent cooperative fusion prediction stage. By leveraging $C_i$ and $S_i$ alongside $Q_i$, we enhance the effectiveness of the cooperative fusion by utilizing richer single-agent predictive outputs.

### 3.2 COOPERATIVE FUSION PREDICTION STAGE

**Information Selection and Sharing.** Most query-based 3D object detection models initialize a large set of object queries to improve query coverage and accelerate model training Liu et al. (2022); Li et al. (2022a); Yan et al. (2023). However, during training, only a small portion of the object queries maintain strong associations with actual target objects, while the majority are predicted as background. These background object queries do not contribute significantly to detection performance yet consume substantial transmission bandwidth when shared among agents. To address this issue, we apply a Top-$k$ strategy to the object queries $Q_i$ based on the object classification scores $S_i$ output from the previous stage. To balance effective fusion with reduced communication costs, we set $k$ to the maximum number of detectable objects by the connected and automated vehicles (CAVs). After filtering, each CAV shares object queries $Q_i$, object centers $C_i$, and object class scores $S_i$. Additionally, the LiDAR poses of the CAVs are shared for subsequent spatial alignment.

**Spatial Alignment and Fusion.** Due to the spatial differences between the CAVs and the ego vehicle, their object queries exhibit significant spatial discrepancies. To solve this issue, we apply the Motion-aware Layer Normalization (MLN) (Wang et al., 2023) to spatially align object queries. Specifically, in our method, we first encode the transformation matrix $E_{cav}^{ego}$ from the CAV to ego vehicle and then applies an affine transformation to $Q_{cav}$. The object centers $C_{cav}$ of the CAVs are transformed into the ego vehicle's coordinate using $E_{cav}^{ego}$. After spatial alignment, we concatenate $Q_{ego}$ and $Q_{cav}$ for further fusion operations: $Q_{all} = Q_{ego} + \sum_i Q_{cav_i}$ To handle the dynamic number of connected vehicles in different V2V scenarios, we set the maximum number of connected vehicles in the system to $L$ and zero-padding the final query to maintain a fixed dimension of $L \times N$.

**Efficient Query Transformer.** After obtaining the object query sequences $Q_{all}$, we input them into our Efficient Query Transformer (EQFormer). EQFormer consists of three query-based self-attention blocks and utilizes the $M_{\text{all}}$ attention mask to enable targeted interactions for object queries. $M_{\text{all}}$ is a combination of three masking mechanisms specifically designed to address the challenges of object query fusion. Further details of the EQFormer are discussed in Section 3.3.

**Cooperative Task Head.** The fused object query sequence $Q_{fused}$, processed by the Efficient Query Transformer, is fed into the task head for 3D bounding box and object class prediction. We normalize the object center sequences $C$ as reference points to accelerate model training. Then, a bipartite matching algorithm Carion et al. (2020) is applied to assign the predicted results to the ground truths in the manner. The details of the loss function are explained in Section 3.4.

### 3.3 EFFICIENT QUERY TRANSFORMER

To address the challenges in object query fusion, we propose the Efficient Query Transformer (EQFormer), as shown in Fig. 3. Our EQFormer introduces the Integrated Mask $M_{\text{Intergrate}}$, which integrates three distinct masking strategies. The first masking block is Query Selective Mask, which is designed to prevent padded, invalid object queries from interfering with interactions. Then, to mitigate interaction failures caused by significant contextual differences between object queries, we develop the second masking block, Proximity-Constrained Mask, based on object centers, which restricts interactions to spatially proximity object queries. After that, we propose the Score-Selective Mask to focus interactions on object queries that are strongly related to the target, which is developed based on object class scores. Here, we construct the query-based self-attention block by using $M_{\text{Intergrate}}$ as the attention mask in the Multi-Head Self-Attention (MHSA) mechanism, combined with the Feed-Forward Network, EQFormer is built by stacking three query-based self-attention blocks to achieve efficient fusion of the object query sequences.

**Query Selective Mask.** To ensure that only valid object queries participate in interactions, we designed a Query Selective Mask (QSM) mechanism, which masks out zero-padded object queries. The matrix is defined as follows:

$$M_{\text{QSM}}[i,j] = \begin{cases} 0 & \text{if } 0 \le i < AN \text{ and } 0 \le j < AN \\ 1 & \text{otherwise} \end{cases}, \quad M_{\text{QSM}} \in \mathbb{R}^{(L \times N) \times (L \times N)} \tag{1}$$

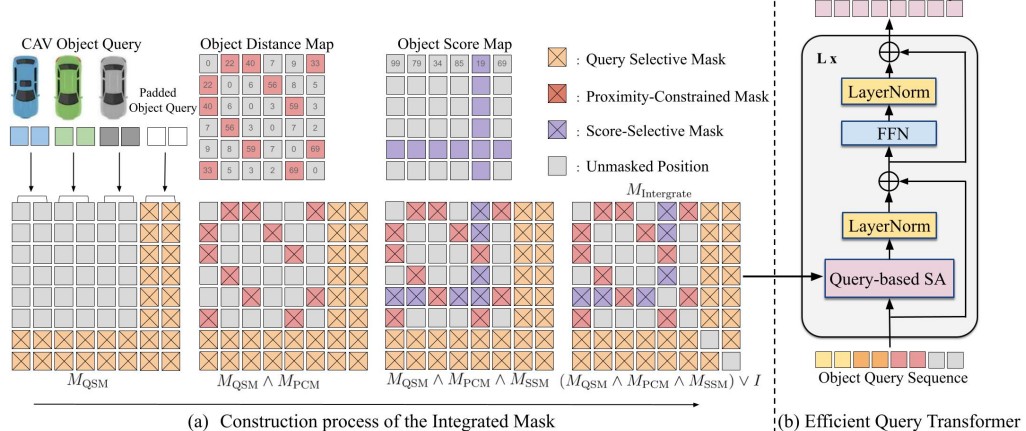

Figure 3: EQFormer architecture. Figure (a) illustrates the construction process of the integrated mask $M_{\text{all}}$. It consists of three mask mechanisms specifically designed to address the challenges of object query fusion: Query Selective Mask, Proximity-Constrained Mask, and Score-Selective Mask. Figure (b) shows the composition of the query-based self-attention block in EQFormer, which contains query-based self-attention equipped with $M_{\text{all}}$ and a feed-forward network (FFN).

Positions beyond $AN$ are assigned a value of 1, indicating masked object queries that are excluded from interactions, ensuring only valid queries are involved.

**Proximity-Constrained Mask.** To ensure that only spatially relevant object queries engage in the fusion stage, we introduce the Proximity-Constrained Mask (PCM). This mechanism limits interactions based on the spatial proximity of the object centers corresponding to each object query. This can potentially cause confusion during feature fusion, when object centers are too far apart, the contextual features between the corresponding queries may vary significantly. To address this, PCM applies a distance threshold $\tau$ to restrict interactions. Specifically, let $C_{all} = \{c_1, c_2, \ldots, c_{L \times N}\}$ represent object centers sequence, where $c_i$ denotes the object center corresponding to the $i$-th object query. We define the spatial distance matrix $D$, with the element $D_{ij}$ representing the Euclidean distance between the $i$-th and $j$-th object centers, formulated as: $D_{ij} = \|c_i - c_j\|$. Based on the matrix $D$ and the distance threshold $\tau$, we introduce the matrix of Proximity-Constrained Mask, expressed as follows:

$$M_{\text{PCM}}[i, j] = \begin{cases} 0, & \text{if } D_{ij} \leq \tau \\ 1, & \text{if } D_{ij} > \tau \end{cases}, \quad M_{\text{PCM}} \in \mathbb{R}^{(L \times N) \times (L \times N)}. \tag{2}$$

Here, the values in the spatial distance matrix exceeds the threshold $\tau$, $M_{\text{PCM}}$, which are set to 1, indicating that the corresponding object queries are masked. Conversely, $M_{\text{PCM}}$ are set to 0, allowing participation in interaction.

**Score-Selective Mask.** In the Information Selection and Sharing module, we employed a Top-k filtering strategy to eliminate most of object queries predicted as background. To further restrict interactions to object queries that are strongly associated with the object targets and improve fusion efficiency, we introduced the Score-Selective Mask, which is an object class score-based masking mechanism. Specifically, let $S_{all} = \{s_1, s_2, \ldots, s_{L \times N}\}$ represent the object class score sequence, where $s_i$ denotes the object class score of the $i$-th object query. Using the confidence threshold $\theta$, the matrix of the Score-Selective Mask is expressed as follows:

$$M_{\text{SSM},i} = \begin{cases} 0, & \text{if } s_i > \theta \\ 1, & \text{if } s_i \leq \theta \end{cases}, \quad M_{\text{SSM}} \in \mathbb{R}^{(L \times N) \times (L \times N)}, \tag{3}$$

where the confidence threshold $\theta$ is set to 0.20, aligning with the threshold used in post-processing. If the object score $s_i$ is less than or equal to $\theta$, $M_{\text{SSM}}$ are set to 1, indicating that the corresponding object query is masked. Conversely, $M_{\text{SSM}}$ are set to 0, allowing the corresponding object query to participate in the fusion stage.

**Query-based Self-Attention Block.** We integrate the above three object query interaction mechanisms into a unified mask, termed $M_{\text{all}}$, which serves as the Attention Mask input for the self-attention block. This self-attention block and feed-forward network (FFN), form our query-based self-attention block. These operations are formulated as follows:

$$M_{\text{all}} = (M_{\text{QSM}} \wedge M_{\text{PCM}} \wedge M_{\text{SSM}}) \vee I, \tag{4}$$

$$\text{Attention}(Q, K, V, M_{\text{all}}) = \text{softmax}\left(\frac{QK^T}{\sqrt{d_k}} + M_{\text{all}}\right)V, \tag{5}$$

$$Q_{\text{fused}} = \text{EQFormer}(Q_{\text{all}}, M_{\text{intergrate}}). \tag{6}$$

The object query sequences $Q_{\text{all}}$ are fed into the EQFormer, achieving efficient fusion of object queries from different CAVs, and output the fused object queries $Q_{\text{fused}}$.

## 3.4 SYNERGISTIC DEEP SUPERVISION

In current cooperative perception systems, improving the accuracy of a single agent's perception enhances the overall performance of the cooperative perception. This implies a positive reinforcement between the Single-Agent Independent Prediction and the Cooperative Fusion Prediction Stages. To achieve that, we introduce a Synergistic Deep Supervision approach and apply deep supervision to both stages simultaneously. During the Single-Agent independent prediction stage, $Q_{\text{ego}}(i)$ from each layer of the ego vehicle's decoder is fed into the Single-TaskHeads. In the collaborative fusion prediction stage, $Q_{\text{fused}}(j)$ from each layer of the EQFormer is fed into the Co-TaskHeads for regression and classification prediction. These operations are formulated as follows:

$$\hat{r}_{\text{single}}(i), \hat{c}_{\text{single}}(i) = \text{Single-TaskHeads}(Q_{\text{ego}}(i)), \tag{7}$$

$$\hat{r}_{\text{co}}(j), \hat{c}_{\text{co}}(j) = \text{Co-TaskHeads}(Q_{\text{fused}}(j)), \tag{8}$$

where $\hat{r}_{\text{single}}(i)$ and $\hat{r}_{\text{co}}(j)$ represent the regression predictions at each stage, while $\hat{c}_{\text{single}}(i)$ and $\hat{c}_{\text{co}}(j)$ denote the classification predictions.

In our method, we utilize identical loss functions for both stages. The classification loss is based on Cross-Entropy Loss, and the regression loss employs $L_1$ Loss. The loss functions are defined as:

$$L_{\text{single}} = \sum_{i=1}^{I} \left(w_1 L_{\text{reg}}(r_{\text{single}}(i), \hat{r}_{\text{single}}(i)) + w_2 L_{\text{cls}}(c_{\text{single}}(i), \hat{c}_{\text{single}}(i))\right), \tag{9}$$

$$L_{\text{co}} = \sum_{j=1}^{J} \left(w_1' L_{\text{reg}}(r_{\text{co}}(j), \hat{r}_{\text{co}}(j)) + w_2' L_{\text{cls}}(c_{\text{co}}(j), \hat{c}_{\text{co}}(j))\right), \tag{10}$$

where $w_1$, $w_2$, and $w_1'$, $w_2'$ are the weights controlling the regression and classification losses in the two stages. Deep supervision is applied in both stages to facilitate faster model convergence. Therefore, our final loss function is:

$$L = w_{\text{single}} L_{\text{single}} + w_{\text{co}} L_{\text{co}}, \tag{11}$$

where $w_{\text{single}}$ and $w_{\text{co}}$ are weights that balance the contributions of the losses from the two stages.

## 4 EXPERIMENTS

### 4.1 DATASETS AND EVALUATION

**Datasets.** We conducted extensive experiments on two multi-agent datasets: OPV2V (Xu et al., 2022c) and V2V4Real (Xu et al., 2023b). OPV2V (Xu et al., 2022c) is a large-scale, multi-modal simulated V2V perception dataset. The train/validation/test splits are 6,694/1,920/2,833, respectively. V2V4Real (Xu et al., 2023b) is an extensive real-world cooperative V2V perception dataset, which is split into 14,210/2,000/3,986 frames for training, validation, and testing, respectively.

**Evaluation.** Following (Xiang et al., 2023), we evaluate three primary settings on this dataset: 1) Homogeneous camera-based detection (V2V-C), 2) Homogeneous LiDAR-based detection (V2V-L), and 3) Heterogeneous camera-LiDAR detection (V2V-H). We adopt Average Precision (AP) at Intersection-over-Union (IoU) thresholds of 0.5 and 0.7 to evaluate the model performance. The communication range between agents is set to 70m.

Table 1: Main performance and bandwidth comparison on OPV2V and V2V4Real Dataset. To further enhance model performance, we expanded the detection range of HMViT, PyramidFusion, and CoCMT to $[-102.4m, +102.4m]$ in the V2V-C setting of the OPV2V dataset. For CoCMT, we transmits the Topk(k=50) object queries during inference.

| Dataset | OPV2V | | | | | | V2V4Real | | |
| --- | --- | --- | --- | --- | --- | --- | --- | --- | --- |
| Setting | V2V-C | | V2V-L | | V2V-H | | V2V-L | | Bandwidth (Mb) |
| Metric | AP50↑ | AP70↑ | AP50↑ | AP70↑ | AP50↑ | AP70↑ | AP50↑ | AP70↑ | |
| AttFusion | 0.447 | 0.184 | 0.895 | 0.779 | 0.624 | 0.411 | 0.701 | 0.454 | 536.8 |
| CoBEVT | 0.466 | 0.168 | 0.933 | 0.823 | 0.811 | 0.504 | 0.684 | 0.404 | 134.2 |
| V2X-ViT | 0.518 | 0.259 | 0.940 | 0.830 | 0.858 | 0.667 | 0.659 | 0.426 | 134.2 |
| HM-ViT | 0.523 | 0.278 | 0.947 | 0.861 | 0.861 | 0.699 | 0.419 | 0.419 | 134.2 |
| PyramidFusion | 0.634 | 0.412 | 0.957 | **0.921** | 0.842 | 0.765 | 0.712 | 0.460 | 134.2 |
| CoCMT (Late) | 0.611 | 0.385 | 0.969 | 0.894 | 0.817 | 0.621 | 0.693 | 0.418 | 0.024 |
| CoCMT (Interm) | **0.634** | **0.445** | **0.971** | 0.911 | **0.879** | **0.771** | 0.710 | **0.471** | 0.416 |

## 4.2 EXPERIMENTAL SETUPS

**Implementation Details.** we employ the query-based 3D detection model, CMT (Yan et al., 2023), as the primary model in the single-agent stage. For the Camera agent, we employ the CMT-C variant, which utilizes ResNet-50 as the camera encoder. For the LiDAR agent, we employ the CMT-L variant, which utilizes PointPillar as the LiDAR encoder. SPCONV2 (Contributors, 2022) is applied for voxelization of the point cloud data. In both stages, all feature dimensions are set to 256, including point cloud tokens, image tokens, and object queries.

**Training strategy.** For V2V-L, we adopt the training strategy described in Section 3.4, We utilize a Top-$k$ selection strategy to transmit 120 object queries ($k = 120$). For V2V-H, we load the single-agent model (CMT-C) weights along with the multi-agent model weights trained in the V2V-L scenario. The Top-$k$ selection strategy is applied to transmit $k = 120$ object queries. For V2V-C, we train the model in an end-to-end manner, transmitting all 900 object queries.

**Compared Methods.** We adopt the late fusion method from the single-agent model of our framework as the baseline, which aggregates detection results from all CAVs and generates the final output. For the intermediate fusion methods, we benchmark five SOTA methods: ATTFuse (Xu et al., 2022c), CoBEVT (Xu et al., 2022a), V2X-ViT (Xu et al., 2022b), HMViT (Xiang et al., 2023), and HEAL (PyramidFusion) (Lu et al., 2024). These approaches all use feature maps as the medium for information exchange and employ LSS (Philion & Fidler, 2020) to construct BEV features for camera branch. In our experiments, ResNet50 and PointPillar served as the backbone networks for the camera and LiDAR branches, respectively.

## 4.3 QUANTITATIVE EVALUATION

**Perception performance and bandwidth.** Figure 1 demonstrates the trend of AP70 as a function of bandwidth on the OPV2V dataset. Under the V2V-L, V2V-C, and V2V-H settings, at the same bandwidth, our object-query-based model CoCMT significantly outperforms the feature-map-based intermediate fusion models. Additionally, as the bandwidth decreases, the performance degradation of the CoCMT is considerably smaller compared to the feature-map-based model, highlighting the transmission efficiency of object query and their adaptability to bandwidth limitations. Table. 1 presents a performance comparison on the OPV2V and V2V4Real datasets. Our proposed CoCMT model transmits only the Top-$k$ ($k = 50$) object queries during inference, requiring just **0.416** Mb of bandwidth, which reduces bandwidth consumption by **323x** compared to the feature-map-based SOTA intermediate fusion model. Despite the significant reduction in bandwidth, CoCMT still demonstrates excellent performance across multiple settings: on the OPV2V dataset, AP70 outperforms the SOTA intermediate fusion model by 2.7 and 0.6 points in the V2V-C and V2V-H settings, respectively; AP50 improves by 1.4 points in the V2V-L setting; and AP70 increases by 1.1 points in the V2V-L setting of the V2V4Real dataset. This indicates that CoCMT not only offers significant transmission efficiency but also maintains superior performance in low-bandwidth environments. Furthermore, CoCMT's intermediate fusion method significantly outperforms the single-agent late fusion method, particularly on the V2V4Real dataset, where AP70 and AP50 are

improved by 5.3 and 1.7 points, respectively. This further highlights the performance advantages of our object-query-based intermediate fusion method.

**Efficient Inference Experiment.** Figure. 4 demonstrates the performance variation of our model when reducing transmission bandwidth during inference. Our model employs a class score-based Top-$k$ strategy during inference to reduce the number of transmitted object query, thereby lowering transmission bandwidth. When the number of transmitted object query is reduced from 120 to 30, model performance remains nearly unaffected. Only when the transmission is reduced to 20, a slight performance drop is observed in the V2V-H and V2V-L settings. This indicates that our object score mask effectively limits interactions to only strongly related object query.

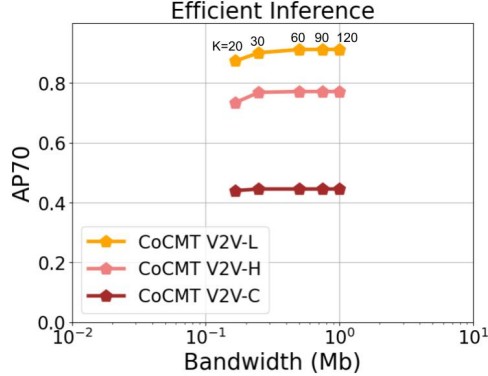

Figure 4: Top-$k$ selection strategy at inference.

## 4.4 ALATION STUDY

**Component Ablation Study.** We conducted ablation experiments on the core design of CoCMT, with results shown in Table 2. The results indicate that each design component significantly enhances model performance. First, the Query Selective Mask $M_{QSM}$ filters out padded zero-value queries, preventing them from interfering with the fusion process and en-

Table 2: Components ablation studies.

| EQFormer | $M_{QSM}$ | $M_{PCM}$ | $M_{SSM}$ | AP50 ↑ | AP70 ↑ |
|---|---|---|---|---|---|
| | | | | 0.622 | 0.345 |
| ✓ | | | | 0.691 | 0.437 |
| ✓ | ✓ | | | 0.691 | 0.440 |
| ✓ | ✓ | ✓ | | 0.721 | 0.465 |
| ✓ | ✓ | ✓ | ✓ | 0.710 | 0.471 |

suring model stability. Second, the Proximity-Constrained Mask $M_{PCM}$ restricts object query interactions to spatially adjacent areas, enabling efficient and accurate fusion within a reasonable spatial range. Lastly, the Score-Selective Mask $M_{SSM}$ further improves the focus of the fusion process by limiting interactions to only those object queries highly relevant to the target. Combining these three masking mechanisms allows EQFormer to fuse object queries effectively for optimal performance.

**Proximity-Constrained Mask Distance Ablation.** The distance threshold in the Proximity-Constrained Mask directly influences the interaction range between object queries, which in turn has a significant impact on model performance. In Table 3, we conducted an ablation study to evaluate the effects of different threshold values. When the threshold is set to infinity, meaning no proximity-constrained restrictions are applied to interactions between object queries (i.e., the Proximity-Constrained Mask is not used), the model's performance sig-

Table 3: $M_{PCM}$ distance ablation.

| $M_{PCM}$ | AP50 ↑ | AP70 ↑ |
|---|---|---|
| $+\infty$ | 0.690 | 0.419 |
| 30m | 0.696 | 0.440 |
| 20m | 0.700 | 0.452 |
| 10m | 0.710 | 0.471 |
| 5m | 0.683 | 0.430 |

nificantly declines. We believe this is due to the large contextual differences between object queries, which lead to failed feature fusion. In contrast, when the distance threshold is set to 10 meters, the model achieves optimal performance. Although increasing the threshold further expands the interaction range, it also introduces unreasonable interactions between object queries that are too far apart, ultimately resulting in reduced model performance. This demonstrates that the Proximity-Constrained Mask plays a key role in improving model performance by effectively controlling the interaction range between object queries.

## 4.5 QUALITATIVE EVALUATION

**Detection visualization.** Figure 5 presents the detection visualizations of CoCMT and Pyramid-Fusion on the OPV2V and V2V4Real datasets. As shown in the V2V-C setting of OPV2V, our CoCMT achieves higher detection accuracy, with predicted bounding boxes showing a greater overlap with ground truths. In the V2V-L setting of both OPV2V and V2V4Real dataset, CoCMT detects more dynamic objects, showcasing the efficiency of using object query as a medium for information

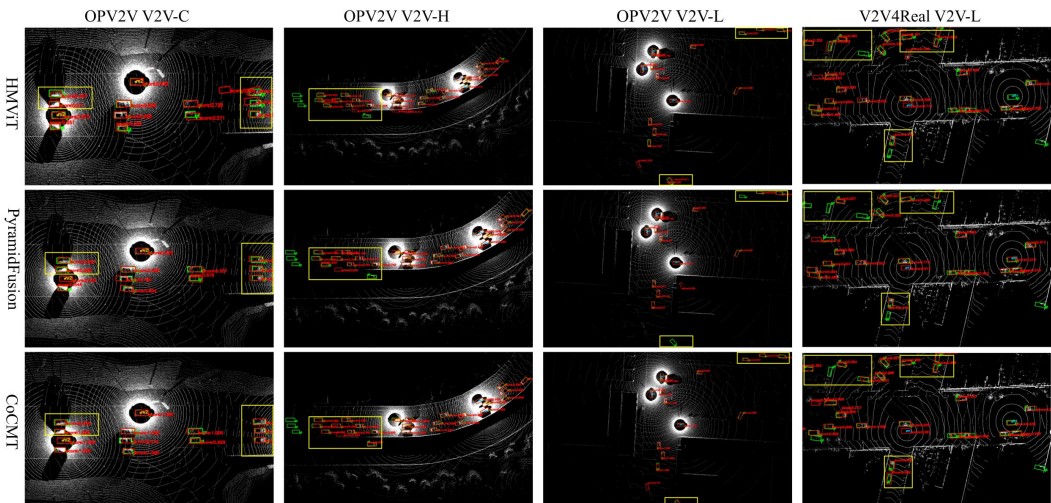

Figure 5: **Qualitative visualizations on the OPV2V and V2V4Real datasets.** Green and red 3D bounding boxes represent the ground truth and predictions, respectively. Key areas are highlighted with yellow boxes. Our method provides more accurate detection results and identifies more targets. Additional visualizations are provided in the supplementary materials.

transmission. In the V2V-H setting of OPV2V, CoCMT also achieves higher accuracy and broader detection coverage within the detection range of connected camera agents, demonstrating that our approach can effectively handle both homogeneous and heterogeneous multi-agent perception tasks through a unified and concise architecture.

## 5 CONCLUSION

In this paper, we introduce the CoCMT framework to address the challenges of collaborative perception in both homogeneous and heterogeneous multi-agent environments. By utilizing object queries as the medium for information transmission, the framework significantly reduces bandwidth consumption while enhancing the efficiency of collaborative perception. The Efficient Query Transformer (EQFormer) is designed with three masking mechanisms to precisely regulate interactions between object queries, ensuring focused and efficient fusion. Additionally, the Synergistic Deep Supervision mechanism applies deep supervision across both stages, accelerating model training. Extensive experiments on both simulated and real-world datasets validate the bandwidth efficiency of our proposed CoCMT framework, demonstrating its capability to achieve superior performance compared to state-of-the-art methods with orders-of-magnitude bandwidth savings. We hope our work will facilitate resource-constraint, communication-efficient collaborative perception frameworks towards safer, more robust mobility systems.

**Limitations.** The single-agent model in our framework uses a DETR-based architecture. Compared to anchor-based models, DETR-based models converge slowly and require higher training costs. We could consider using the query denoising methods mentioned in (Li et al., 2022a; Zhang et al., 2022; Wang et al., 2023) to accelerate the model training. Additionally, our model is also suitable for multi-modal cooperative perception, where each agent simultaneously uses both LiDAR and camera sensors. In our future work, we plan to explore this capability further using real-world multimodal cooperative perception datasets.

**Broader Impact.** Our proposed CoCMT framework has the potential to significantly advance the field of autonomous driving by improving the efficiency and scalability of cooperative perception systems. However, the deployment of such systems also raises important considerations. First, sharing information among vehicles involves transmitting sensitive information. To protect against potential cyber-attacks or data breaches, robust encryption, and secure communication protocols must be implemented. Furthermore, the increased reliance on automated systems may impact employment in the transportation sector and raise questions about accountability in the event of system failures. In the future, researchers and engineers should handle these challenges responsibly.

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

## A APPENDIX

### A.1 MULTI RANGE LABEL SELECTION

We adopted a multi-range label selection method and constructed corresponding ground truth for two stages: $r_{single}$ and $c_{single}$ for the single-agent independent prediction stage, and $r_{co}$ and $c_{co}$ for the cooperative fusion prediction stage. This strategy offers several advantages: not only does it expand the cooperative perception detection range under the V2V-C setting, but it also reduces the learning complexity during the cooperative fusion stage and effectively addresses challenges posed by differing detection ranges of heterogeneous sensors in the V2V-H setting. We configured the detection ranges and ground truth for the three cooperative perception settings: V2V-L, V2V-C, and V2V-H. Using the OPV2V dataset as an example, the selection results are shown in Table. 4.

Table 4: Specific Configuration Settings

| Setting | Ego Detection and GT Range (m) | Collaborative Detection and GT Range (m) |
|---------|-------------------------------|------------------------------------------|
| V2V-L | $[-102.4, -102.4, +102.4, +102.4]$ | $[-102.4, -102.4, +102.4, +102.4]$ |
| V2V-C | $[-51.2, -51.2, +51.2, +51.2]$ | $[-102.4, -102.4, +102.4, +102.4]$ |
| V2V-H | L: $[-102.4, -102.4, +102.4, +102.4]$
C: $[-51.2, -51.2, +51.2, +51.2]$ | $[-102.4, -102.4, +102.4, +102.4]$ |

**For V2V-C Setting**: Unlike most cooperative perception methods Xiang et al. (2023); Lu et al. (2024); Xu et al. (2022a) that use a detection range of only $51.2m$, we maintained the camera's detection range and ground truth of $51.2m$ in the single-agent independent prediction stage, while extending the detection range to $102.4m$ during the cooperative fusion stage. Through a cooperative deep supervision mechanism, the effective detection range for cooperative perception was successfully expanded.

**For V2V-L Setting**: Due to the larger detection range of the LiDAR, we used a $102.4m$ detection range for both the single-agent prediction and cooperative fusion stages. To improve individual vehicle detection performance, we introduced cooperative ground truth in the single-agent stage, increasing the number of prediction labels, thereby reducing the difficulty of subsequent cooperative fusion.

**For V2V-H Setting**: In the OPV2V Xu et al. (2022c) dataset, the camera's effective detection range is $51.2m$, while the LiDAR's is $102.4m$. Unlike HMViT Xiang et al. (2023), which simplifies heterogeneous feature fusion by unifying the detection range to $102.4m$, our framework flexibly handles differences in detection ranges between heterogeneous sensors. In the single-agent independent prediction stage, each sensor used its effective detection range and ground truth. During the cooperative fusion prediction stage, we unified the detection range to $102.4m$, leveraging the cooperative ground truth to further improve the accuracy of individual vehicle predictions.

### A.2 DETECTION VISUALIZATION

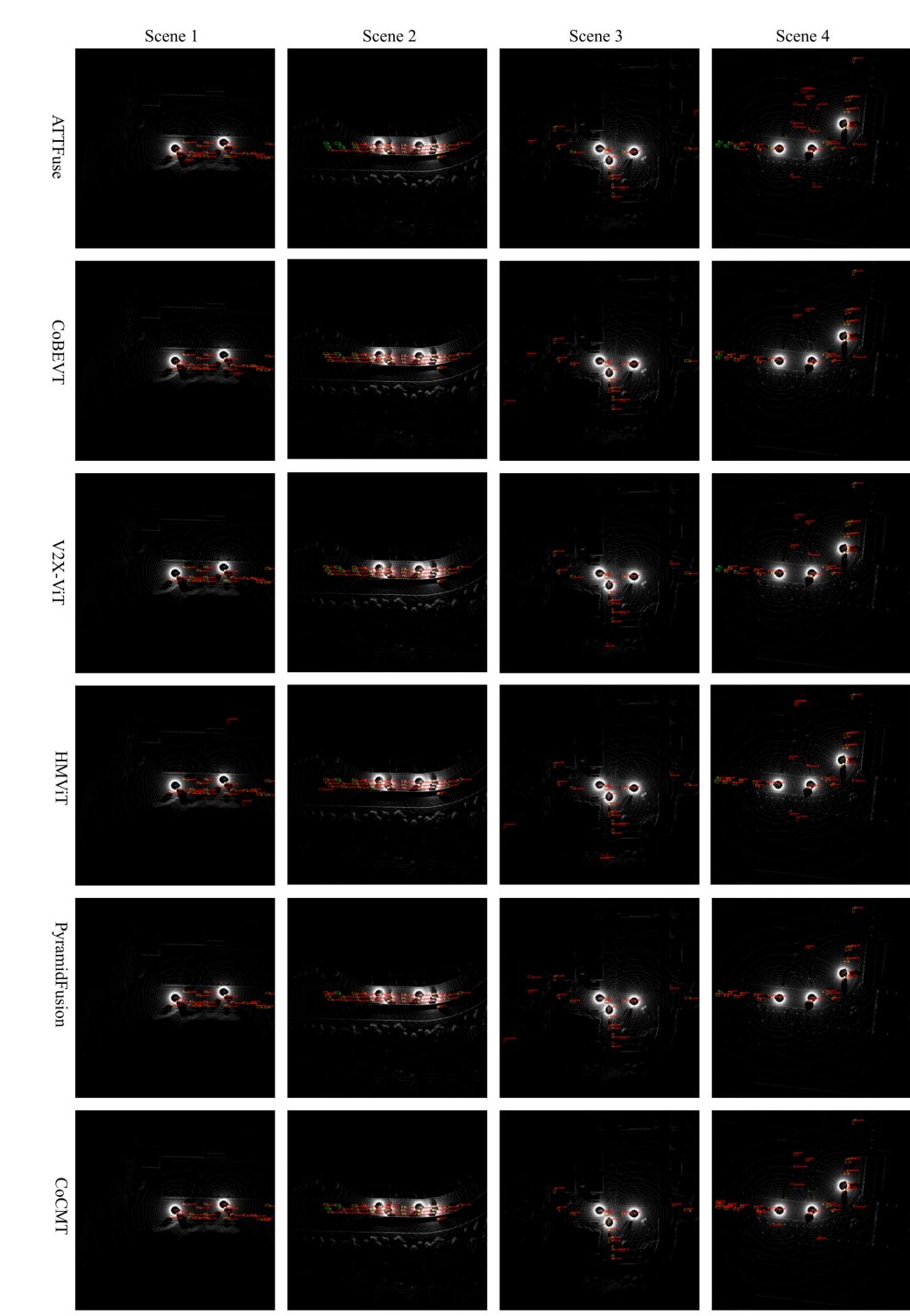

Figure 6: **Qualitative comparison on scenarios 1-4 under V2V-L setting in the OPV2V dataset.** The green and red bounding boxes represent the ground truth and prediction, respectively. Our method detected more dynamic objects.

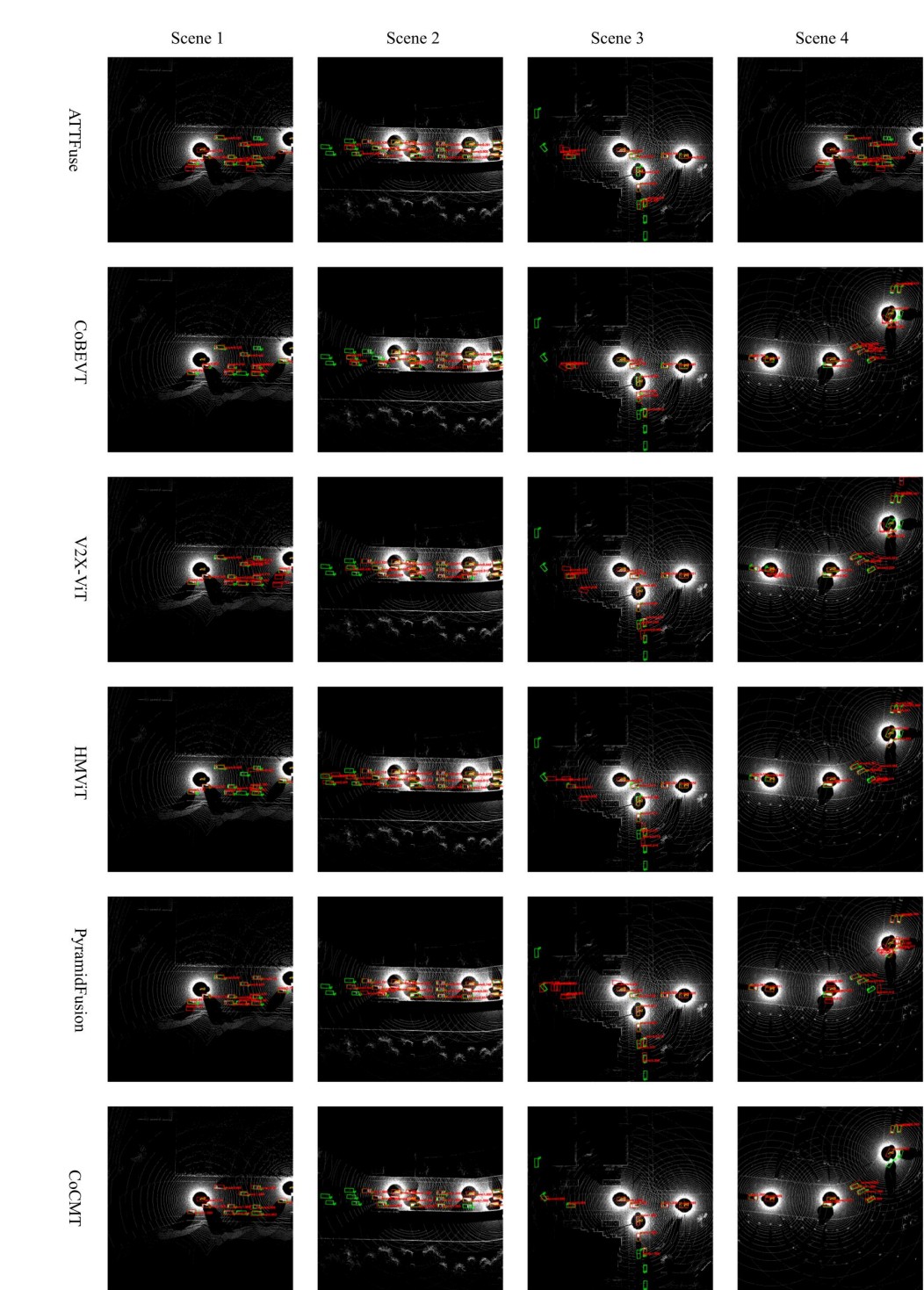

Figure 7: **Qualitative comparison on scenarios 1-4 under V2V-C setting in the OPV2V dataset.** The green and red bounding boxes represent the ground truth and prediction, respectively. Our method produced more accurate detection results.

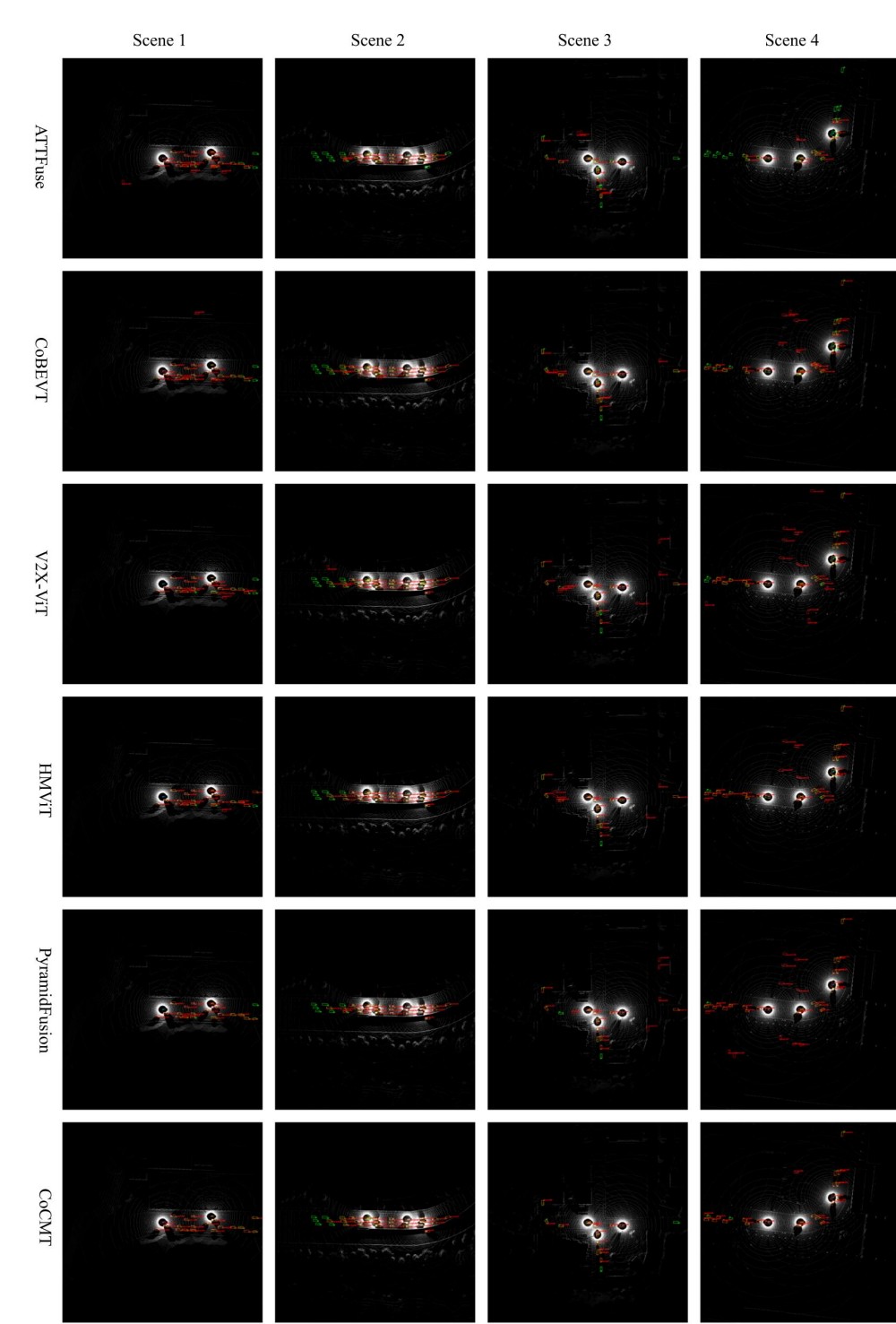

Figure 8: **Qualitative comparison on scenarios 1-4 under V2V-H setting in the OPV2V dataset.** The green and red bounding boxes represent the ground truth and predictions, respectively. Our method produced more accurate detection results and resulted in fewer false detection boxes.

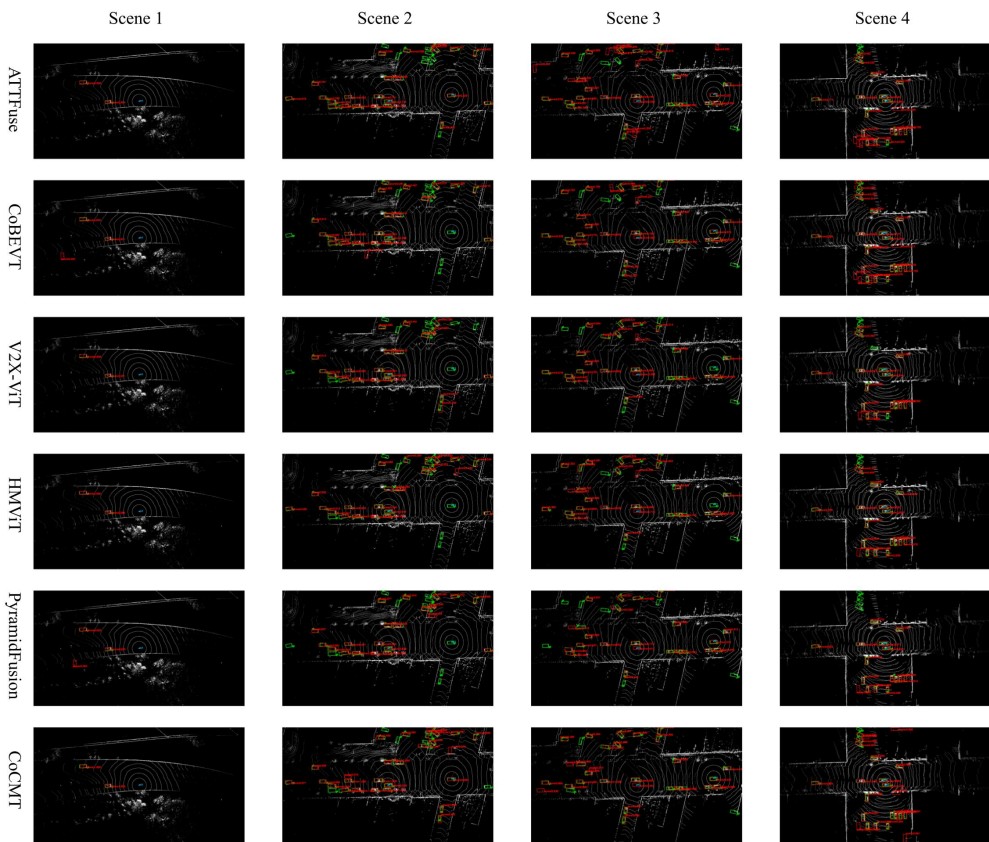

Figure 9: **Qualitative comparison on scenarios 1-4 in the V2V4Real dataset.** The green and red bounding boxes represent the ground truth and predictions, respectively. Our method produced more accurate detection results.

