# OpenReview forum: "CoCMT: Towards Communication-Efficient Corss-Modal Transformer For Collaborative Perception"
_ICLR.cc/2025/Conference — ICLR 2025 Conference Withdrawn Submission_

### Official Review · Reviewer_GWf6 · 2024-11-03

**Soundness:** 2
**Presentation:** 2
**Contribution:** 2
**Rating:** 3
**Confidence:** 5

**Summary:**

This paper presents CoCMT, an object-query-based collaboration framework that enables efficient communication while unifying homogeneous and heterogeneous cooperative perception tasks. They design the Efficient Query Transformer (EQFormer), which incorporates three masking mechanisms to limit interactions between object queries to spatially valid, proximate, and strongly target-associated areas, ensuring precise and efficient attention learning for fusion. They introduce a Synergistic Deep Supervision mechanism that applies deep supervision at both the individual prediction and collaborative fusion stages.

**Strengths:**

1. This paper is clearly structured and written-well. Proper formulization makes the processes of the method easy to understand.
2. This paper focuses on an interesting issue, namely that most existing cooperative systems transmit feature maps, such as bird’s-eye view (BEV) representations, which include substantial background data and are costly to process due to their high dimensionality.

**Weaknesses:**

1. The proposed method lacks validation experiments on some mainstream datasets, such as V2XSim and DAIR-V2X.
2. The proposed collaborative perception method has only been validated on the 3D detection task, lacking validation on other tasks such as segmentation and trajectory prediction.
3. The proposed method as a whole is a combination improvement of query-based method QUEST and multimodal 3D detection method CMT, which to some extent weakens the innovation of the proposed method.
4. Lack of detailed ablation experiments on some hyperparameters, such as the confidence threshold in Score Selective Mask, weights that balance the contributions of the losses from the two stages.
5. The paper claims that the Synergistic Deep Supervision mechanism can accelerate model convergence during training and enhance overall performance, but it lacks corresponding experimental validation. Moreover, the ablation study section does not include a detailed ablation experiment for the Synergistic Deep Supervision component.
6. Table 1 and Figure 1 lack comparison with some existing communication-efficient collaborative perception methods, such as Where2comm, HEAL, CodeFilling, QUEST, etc. Of course, there are many more.

**Questions:**

1. Please refer to the Paper Weaknesses mentioned above.
2. Why does AP50 decrease after adding Score Selective Mask in Table 2? Please analyze the reasons.
3. Figures 5, 6, 7, 8, and 9 become severely distorted and very blurry when enlarged, making it difficult to see clearly. Please use vector graphics for visualization to clearly present the detection results.
4. The proposed method is only applicable to DETR-based 3D detectors and cannot be extended to anchor-based models.

---

### Official Review · Reviewer_dBbU · 2024-11-04

**Soundness:** 2
**Presentation:** 3
**Contribution:** 1
**Rating:** 1
**Confidence:** 5

**Summary:**

As claimed in the introduction, the paper aims to resolve the two primary limitations in the existing works. On the one hand, the work focuses on achieving the balance between communication efficiency and perception performance. On the other hand, the work addresses the heterogeneity of sensor systems among vehicles. To this end, the work proposed a query-based Transformer architecture and a deep supervision training strategy. The core component, EQFormer, detects and extracts highly interactive areas to ensure precise and efficient attention while reducing the bandwidth

**Strengths:**

The paper is well organized overall, with clear statements on objectives and corresponding design logic to address the issues. The experiment section of the paper is fairly detailed, with quantitative results and qualitative presentations. The ablation study, at its bottom line, serves to justify the correctness of the design.

**Weaknesses:**

The paper has severe issues with its fundamental design logic and comparative experiment designs. Here are a few critical problems:
1. In section 2.2, the authors refer to existing works such as [Xu, 2022] and [Xu, 2024] and state that these methods only investigated camera-only homogeneous perception. However, these papers investigated the combination of other modalities, such as LIDAR, with the camera. This shows that the authors have never genuinely read these papers and made careless assertions. Similar issues can be found in other references in the paper.
2. The authors identify that feature maps such as BEV features are the major issue concerning communication bandwidth and computation overhead. However, existing works have explored the use of BEV features to overcome these limitations. For example, in the paper "MACP: Efficient Model Adaptation for Cooperative Perception," from WACV 2024, the authors use convolution layers to achieve information compression and comparative performance on the same two benchmark datasets of this work. The authors would need to compare their method to this existing work as well as other prior works in this area that address the same issue to showcase that CoCMT with EQFormer has promised improvements.
3. The heterogeneity of the sensor system is a challenge the authors brought up several times in the paper. However, neither in the methodology nor in the experiment result can we find strong evidence to support that the proposed method has a specific design targeting this specific issue. Together with the misleading assertion about prior works, the authors have showcased no contribution to addressing the issue.

**Questions:**

Questions have been raised in the discussion about the weakness of this paper. Here listed them in short:
1. Why do the authors ignore the fact that many of their cited works have, in fact, explored heterogeneous sensor systems?
2. If communication bandwidth and computation are the main concerns, what justifies CoCMT with EQFormer as a better alternative to the existing methods?
3. The authors assert that high-dimensional feature maps are the root cause of communication and computation overheads and are unnecessary for perception. Can you evaluate your proposed method operating on the feature maps and showcase a worse or similar performance to fairly justify feature maps not being needed?

---

### Official Review · Reviewer_6t9x · 2024-11-04

**Soundness:** 2
**Presentation:** 3
**Contribution:** 2
**Rating:** 6
**Confidence:** 4

**Summary:**

This paper focuses on the communication bottleneck of cooperation perception systems for efficient, economical and robust cooperation in sharing visual information. It points out the ignorance of communication cost of homogeneous multi-agent perception in real-world scenarios, and proposes query-based model: CoCMT to aggregate and transmit numerous, scattered and unordered individual queries about visual objects. For the queries selection, the author proposed three masking mechanisms to ensure the target-associated fusion. For the cooperation efficiency, the author designed an two-step supervision mechanism to accelerates model convergence. All the contributions are proofed significant in experiments.

**Strengths:**

1.	The author has skillfully synthesized sources, providing a comprehensive overview of the current state of research on cooperative perception fusion and 3D object detection methods. The critical analysis of the existing literature is insightful and well-structured, revealing a deep understanding of the topic.
2.	The author proposed an efficient query Transformer to selection object related queries. They used three masking strategies to identify query which is valid, strong object relevant and better performance in prediction to achieve feature fusion.
3.	The author has built up a two-stage CoCMT framework and designed a synergistic deep supervision mechanism to achieve individual prediction and cooperation with sparse information aggregation.
4.	This paper proposes a reasonable solution with well-developed and reliable experiments. The author compared CoCMT with five state-of-the-art solutions on two real-world datasets, the results highlight the information compression ratio and communication efficiency with sound 3-D detection performance. It also builds up ablation studies to illustrate the efficiency of masking mechanisms and discusses the impact of threshold on model performance.

**Weaknesses:**

Some proposals are not very solid. For example, the author mentions they remain “the task heads” in single-agent independent prediction stage to “allow us to incorporate additional predictive information”. But this paper didn’t explain it in detail, as how to use it, why it has this advantage, why we need this strategy. The author also introduces a supervision mechanism as a two-part loss function to accelerate the model training, but failed to proof the improvement of adding this loss into the model. This paper needs some theoretical or actual evidences as convergence analysis or ablation study.

**Questions:**

The key point of efficient query Transformer seems using a sparse matrix to control what information we need to attention, as the author said, the threshold of each masking mechanism is significant. So how to choose the best threshold to make sure the query match the target object closely?

---

### Official Review · Reviewer_jKys · 2024-11-11

**Soundness:** 3
**Presentation:** 3
**Contribution:** 2
**Rating:** 5
**Confidence:** 4

**Summary:**

This paper presents CoCMT, a novel framework for communication-efficient cooperative perception tailored to autonomous vehicles. By using object queries rather than traditional high-dimensional feature maps, CoCMT significantly reduces bandwidth requirements while enhancing perception performance. The framework integrates homogeneous and heterogeneous multi-agent perception tasks in a unified manner. The proposed Efficient Query Transformer (EQFormer) employs innovative masking strategies to address query sparsity and unordered interactions, ensuring precise and efficient fusion. Additionally, a Synergistic Deep Supervision mechanism accelerates training by reinforcing predictions across both single-agent and cooperative stages. Experimental results on OPV2V and V2V4Real datasets demonstrate superior performance and bandwidth efficiency, achieving a 323x reduction in bandwidth compared to state-of-the-art methods while maintaining or improving perception accuracy. The paper also provides extensive quantitative and qualitative evaluations, highlighting the advantages of the framework in real-world scenarios.

**Strengths:**

1. CoCMT's object query-based approach is a compelling alternative to traditional feature map-based methods, addressing bandwidth and complexity trade-offs effectively.

2. The design of EQFormer with its masking mechanisms enhances query fusion efficiency and accuracy.

3. Extensive experiments validate the framework's advantages in performance and bandwidth efficiency.

4. The approach supports heterogeneous sensor data, expanding its applicability to various multi-agent perception tasks.

**Weaknesses:**

The novelty of using object queries as intermediaries for communication is interesting but could be perceived as incremental. Previous works, such as QUEST [Fan et al., 2024], have explored query-based approaches, and the paper does not sufficiently distinguish its contributions from these prior efforts.

While the three masking mechanisms in EQFormer are an improvement, they primarily represent refinements of standard attention techniques rather than introducing a fundamentally new paradigm. The work appears to be more focused on engineering optimizations tailored for collaborative perception rather than advancing foundational machine learning theory.

The framework's reliance on established components, such as DETR-based object detection and standard feature fusion techniques, raises questions about whether the contributions lie more in implementation than in conceptual innovation.

The framework is narrowly tailored to collaborative perception for autonomous vehicles, specifically in V2X scenarios. While this is a critical application, the lack of broader applicability limits the relevance of the work to the general machine learning community. The methods and findings may not translate well to domains beyond multi-agent perception systems in autonomous driving.

The paper could benefit from a more comprehensive literature review, as some highly relevant works about collaborative perception are missing [1-6]. Including a broader range of recent studies would provide a stronger context for the contributions and better situate the proposed framework within the current state of research.

[1] Li, Y., Ren, S., Wu, P., Chen, S., Feng, C. and Zhang, W., 2021. Learning distilled collaboration graph for multi-agent perception. Advances in Neural Information Processing Systems, 34, pp.29541-29552.

[2] Li, Y., Ma, D., An, Z., Wang, Z., Zhong, Y., Chen, S. and Feng, C., 2022. V2X-Sim: Multi-agent collaborative perception dataset and benchmark for autonomous driving. IEEE Robotics and Automation Letters, 7(4), pp.10914-10921.

[3] Huang, S., Zhang, J., Li, Y. and Feng, C., 2024. Actformer: Scalable collaborative perception via active queries. ICRA 2024.

[4] Yang, D., Yang, K., Wang, Y., Liu, J., Xu, Z., Yin, R., Zhai, P. and Zhang, L., 2024. How2comm: Communication-efficient and collaboration-pragmatic multi-agent perception. Advances in Neural Information Processing Systems, 36.

[5] Su, S., Li, Y., He, S., Han, S., Feng, C., Ding, C. and Miao, F., 2023, May. Uncertainty quantification of collaborative detection for self-driving. In 2023 IEEE International Conference on Robotics and Automation (ICRA) (pp. 5588-5594). IEEE.

[6] Su, S., Han, S., Li, Y., Zhang, Z., Feng, C., Ding, C. and Miao, F., 2024. Collaborative multi-object tracking with conformal uncertainty propagation. IEEE Robotics and Automation Letters.

**Questions:**

What is the key innovation of this paper? While the proposed CoCMT framework effectively combines object queries with communication-efficient mechanisms and introduces the Efficient Query Transformer (EQFormer) with novel masking strategies, these contributions appear to be incremental rather than foundational. Could the authors clarify what distinguishes this work as a conceptual breakthrough in collaborative perception, particularly compared to related methods like QUEST or ActFormer? Additionally, how does this innovation advance the broader field of machine learning or multi-agent systems beyond optimizing for bandwidth efficiency in V2X scenarios? A clear articulation of the conceptual novelty would strengthen the case for this work.

---

### Note · Authors · 2024-11-14

I have read and agree with the venue's withdrawal policy on behalf of myself and my co-authors.